# Mitochondrial DNA Replication and Disease: A Historical Perspective on Molecular Insights and Therapeutic Advances

**DOI:** 10.3390/ijms262110275

**Published:** 2025-10-22

**Authors:** Shruti Somai, Chioma H. Aloh, Dillon E. King, William C. Copeland

**Affiliations:** Genome Integrity and Structural Biology Laboratory, Mitochondrial DNA Replication Group, National Institute of Environmental Health Sciences, National Institutes of Health, Research Triangle Park, Durham, NC 27709, USA; shruti.somai@nih.gov (S.S.); chioma.aloh@nih.gov (C.H.A.); dillon.king@nih.gov (D.E.K.)

**Keywords:** mitochondria, mitochondrial diseases, mtDNA, mtDNA replication, DNA polymerase γ, PolG, PolG2, mtSSB, Twinkle

## Abstract

Mitochondria are vital for cellular energy production, as these organelles generate most of the cellular energy required for various metabolic processes. Mitochondria contain their own circular DNA, which is present in multiple copies and is exclusively maternally inherited. Cellular energy in the form of adenosine 5′-triphosphate is produced via oxidative phosphorylation and involves the coordinated expression of genes encoded by both the nuclear and mitochondrial genomes. Mitochondrial DNA itself is replicated by a dedicated set of nuclear-encoded proteins composed of the DNA polymerase gamma, the Twinkle helicase, the mitochondrial single-stranded DNA binding protein, as well as several accessory factors. Mutations in these genes, as well as in the genes involved in nucleotide metabolism, are associated with a spectrum of mitochondrial disorders that can affect individuals from infancy to old age. Additionally, mitochondrial disease can arise as a result of point mutations, deletions, or depletion in the mitochondrial DNA or in genes involved in mitochondrial transcription, replication, maintenance, and repair. Although a cure for mitochondrial diseases is currently elusive, several treatment options have been explored. In this review, we explore the molecular insights of the core mitochondrial replisome proteins that have aided our understanding of mitochondrial diseases and influenced current therapies.

## 1. Introduction

Mitochondria are intracellular organelles present in nearly all eukaryotes. Known mainly for their role in cellular energy production, mitochondria harbor the electron transport chain (ETC), which generates more than 90% of adenosine 5′-triphosphate (ATP) through oxidative phosphorylation (OXPHOS) [1,2,3]. Mitochondria are the major source of endogenous reactive oxygen species (ROS) and are the host of many fundamental biochemical pathways, such as the tricarboxylic acid (TCA) cycle, fatty acid oxidation, as well as certain parts of the urea cycle [4]. In addition to cellular energy production, mitochondria fulfill critical roles in many key biological processes, such as apoptosis, ion homeostasis, iron-sulfur cluster biogenesis, cellular signaling, and immune response regulation [5].

A unique feature of mitochondria is that these organelles harbor their own circular, double-stranded genome, which is exclusively maternally inherited and distinct from the nuclear genome. Human mitochondrial DNA (mtDNA) is 16,569 base pairs (bp) in size and encodes 37 genes, all of which are directly or indirectly involved in ATP production (Figure 1A). Of the 37 genes encoded by mtDNA, 13 genes encode polypeptides that make up essential subunits of the electron transport chain (ETC) (complexes I, III, IV, and V) and are required for oxidative phosphorylation (OXPHOS). The remaining 24 genes encode 22 transfer RNAs and 2 ribosomal RNAs necessary for the mitochondrial translation of the 13 polypeptides [1] (Figure 1A). The OXPHOS system itself consists of five multiprotein complexes and involves more than 90 proteins, encoded by both the nuclear and mitochondrial genomes [6]. The nuclear genome is responsible for encoding most mitochondrial proteins, including the subunits of the ETC complexes, as well as proteins involved in mtDNA maintenance, transcription, replication, and copy number regulation [7]. Although mtDNA encodes only a small proportion of proteins involved in OXPHOS compared to the nuclear genome, these genes are critical for cellular energy production [8,9]. Given that mitochondria are under dual genetic control of both the nuclear and mitochondrial genomes, mutations in either genome may impair ETC function and therefore be detrimental to human health [10,11]. Defects in mitochondrial gene expression significantly impair OXPHOS capacity and lead to a wide spectrum of diseases, which will be further discussed in this review [10,12].

Similar to bacterial chromosomal DNA, mtDNA is organized into nucleoprotein complexes, known as nucleoids, which are located in the inner mitochondrial matrix. Within these nucleoids, mtDNA is compacted by different proteins, with the most abundant being transcription factor A, mitochondrial (TFAM) [13,14]. Human mtDNA consists of two strands, termed heavy strand (H-strand) and light strand (L-strand) due to their strand bias in nucleotide composition. The H-strand has a higher content of guanine and thymine nucleotides compared to the L-strand, allowing these strands to be separated by density gradient centrifugation in alkaline cesium chloride [15,16]. MtDNA also contains a noncoding region (NCR) of 1121 nucleotides that harbors the promoter regions for transcription initiation for both the H-strand and the L-strand, as well as the origin of replication for the H-strand (O_H_) [17] (Figure 1A). Within the NCR, a linear DNA strand exists, referred to as 7S DNA based upon its sedimentation characteristics. The 7S DNA is approximately 650 nucleotides in length and forms a displacement loop (D-loop). Although the exact function of the D-loop is currently unknown, several studies have proposed that the D-loop might play a role in nucleoid organization and may be involved in the regulation of mtDNA replication [18].

Mammalian mtDNA replication differs from nuclear DNA replication in both mechanism and proteins involved [19,20,21]. The mitochondrial replisome consists of a dedicated set of nuclear encoded proteins, with the core components being the mitochondrial RNA polymerase (POLRMT), the replicative mtDNA helicase Twinkle, DNA polymerase γ (pol γ), and the mitochondrial single-stranded DNA-binding protein (mtSSB). Various models of mtDNA replication have been proposed in different cell types and organisms; however, one of the most widely accepted models for mammalian mtDNA replication is the strand-displacement model, which was first proposed in 1972 [22]. According to this model, both strands of the mtDNA are replicated asynchronously and continuously, without the formation of Okazaki fragments [22,23,24,25,26,27] (Figure 1B). Since human mitochondria lack a dedicated DNA primase, POLRMT is responsible for synthesizing short RNA primers required for initiating mtDNA replication. MtDNA replication is first initiated at the H-strand origin of replication (O_H_) located within the NCR [28,29,30]. To initiate the synthesis of the nascent H-strand, the mitochondrial Twinkle helicase first unwinds the double-stranded mtDNA ahead of pol γ, and the displaced parental H-strand is stabilized by mtSSB. H-strand synthesis is performed by the addition of nucleotides to the 3′ end of an existing RNA primer by pol γ [23,24]. Once two-thirds of the nascent H-strand is synthesized, the L-strand origin of replication (O_L_) in the parental H-strand becomes exposed. At this site, a stem-loop structure is formed which is used by POLRMT for initiating primer synthesis for nascent L-strand synthesis. Once primer synthesis has proceeded for approximately 25 nucleotides, POLRMT is replaced by pol γ, allowing for the synthesis of the nascent L-strand in the opposite direction [23,25,31]. At this stage of mtDNA replication, both nascent H- and L-strands are synthesized simultaneously, until two new daughter molecules of mtDNA are formed. After completion of mtDNA replication, the newly formed mtDNA daughter molecules are separated by topoisomerase 3α (TOP3A) [32,33]. Although other mechanisms for mtDNA replication have been proposed, the strand-displacement model of mtDNA replication has received tremendous support, and the validity of this model has been highlighted by findings from biochemical analyses, reconstitution experiments, single-molecule analysis, and in vivo occupancy binding patterns of mitochondrial replisome proteins [19,20,21,22,24,26].

Unlike the nuclear genome, the mitochondrial genome exists in multiple copies per cell, ranging from 100 to 10,000 depending on cellular energy demands, cell type, and differentiation status [34]. Moreover, each cell, as well as each mitochondrion, can contain a mixture of wild-type and mutant mtDNA, a condition referred to as heteroplasmy. The level of heteroplasmy can change throughout development and aging and represents a critical determinant of mitochondrial dysfunction and disease severity [35,36,37,38]. The multicopy nature of mtDNA allows the tolerance of many pathological mutations until both a cell- and mutation-specific threshold is exceeded. This threshold generally varies from 60% to 90% heteroplasmy depending on the specific mutation and cellular context. Beyond this threshold, OXPHOS becomes significantly impaired, resulting in an overall disruption of cellular energy production and cellular dysfunction [39,40].

Despite the high copy number of mtDNA, which naturally serves as a buffer against endogenous and exogenous perturbations, disruption of mtDNA integrity results in a broad spectrum of human diseases, commonly known as mitochondrial diseases [41,42,43,44]. Mitochondrial diseases involving mtDNA defects are characterized by mitochondrial DNA depletion syndrome (MDS), or multiple deletions combined with or without multiple point mutations. MDS is a group of autosomal recessive disorders that manifest with a significant reduction in the mtDNA content within affected organs and tissues [45]. These disorders typically present in early childhood and include Alpers–Huttenlocher syndrome, hepatocerebral syndromes, myocerebrohepatopathy spectrum disorders, fatal myopathies, and Leigh syndrome. In contrast, mtDNA deletions and point mutations are associated with disorders such as progressive external ophthalmoplegia (PEO), ataxia–neuropathy syndromes, as well as rare disorders of TCA cycle abnormalities. Treatment of mitochondrial diseases is extremely challenging given the involvement of multiple organs, disease severity, age of onset, and broad range of symptoms [46,47,48,49,50,51]. Although effective treatments for mitochondrial diseases remain elusive, current treatment strategies involve physical therapy to improve ETC activity combined with dietary changes and providing patients with supplements that enhance ETC function and mitochondrial biogenesis [52,53,54]. Over the past decade, novel therapeutic approaches have emerged and will be discussed in detail later in this review.

Mitochondrial diseases are caused by genetic defects in either the mtDNA or in the nuclear genes of proteins that function in the mitochondria. Genetic alterations can also occur in genes encoding proteins involved in maintaining a balanced mitochondrial nucleotide pool, which cumulatively result in compromised mtDNA integrity, leading to mitochondrial dysfunction [51]. Various studies have estimated that approximately one in every 5000 children and adults will develop a mitochondrial disease sometime in their lifetime [55,56,57]. Mitochondrial diseases are clinically heterogeneous and typically affect multiple organs, in particular those with high energy demands such as the brain, heart, extraocular muscles, kidney, and liver. Adequate amounts of mtDNA are required for the synthesis of essential subunits of the ETC complexes, and thus for cellular energy production. Unsurprisingly, mutations, deletions, and/or depletion of the mtDNA result in cellular dysfunction due to insufficient ATP production to meet the metabolic demands of the affected tissues [54,58,59].

Since there is currently no cure for mitochondrial diseases, biochemical and molecular characterization of mitochondrial replisome proteins remains valuable for elucidating the clinical consequences of specific amino acid variations. In this review, we aim to provide a historical perspective on the molecular insights of the key mitochondrial replisome proteins whose genes have been identified as mitochondrial disease alleles. We focus solely on those genes that are directly involved in mtDNA maintenance, more specifically, the genes that encode the minimal replication machinery. Furthermore, we highlight current therapeutic advances that offer promising treatment strategies for mitochondrial diseases.

## 2. DNA Polymerase γ

Pol γ is the sole replicative polymerase in mitochondria. The subunit structure of eukaryotic pol γ varies across species. In humans and throughout mammals, pol γ exists as a heterotrimer, consisting of a catalytic subunit bound to a dimeric accessory subunit (Figure 2) [60]. The human catalytic subunit is encoded by the *POLG* gene and is also referred to in the literature as the p140 subunit, based on its molecular weight; as PolG when discussing the protein; and by the aliases POLγA and POLG1 [61]. The human accessory subunit is encoded by the *POLG2* gene and is also referred to as the p55 subunit, based on its molecular weight; as PolG2 when discussing the protein; and by the alias POLγB [62]. In *Drosophila*, pol γ forms a heterodimer, with only one accessory subunit attached to the catalytic core [63]. In unicellular yeasts like *Saccharomyces cerevisiae* and *Schizosaccharomyces pombe*, as well as in *Caenorhabditis elegans*, pol γ functions as a single catalytic subunit. This catalytic subunit ranges from 143 kDa in *S. cerevisiae* to 125 kDa in *S. pombe*, the latter being among the smallest known [60].

## 3. Catalytic Subunit, PolG

The catalytic subunit of human pol γ is encoded by the *POLG* gene on chromosome 15q25 and comprises 1239 amino acids with a molecular weight of 140 kDa. Deletion of either *Polg* or *Polg2* in mice results in embryonic lethality by day 8.5, coinciding with a complete loss of mtDNA [64,65]. The p140 subunit exhibits three enzymatic activities: DNA polymerase, 3′→5′ exonuclease, and 5′ dRP lyase. The exonuclease domain is located at the N-terminus, whereas the polymerase active site lies at the C-terminus, separated by a linker region that facilitates interaction with the accessory subunit.

The 5′ dRP lyase activity is implicated in base excision repair, specifically removing the 5′-deoxyribose phosphate left behind by AP endonuclease activity [66]. Pol γ belongs to the family A group of DNA polymerases and shares structural and biochemical similarities with bacteriophage T7 DNA polymerase [61]. The crystal structure of the pol γ–DNA complex has been resolved as a heterotrimer at 3.2 Å, and with bound nucleotides (ddCTP and dCTP) at 3.3 Å resolution [67,68] (Figure 2).

### 3.1. Fidelity of mtDNA Replication

The human pol γ maintains high base substitution fidelity through stringent nucleotide selectivity and intrinsic 3′→5′ exonucleolytic proofreading activity [69]. Pol γ incorporates nucleotides with high accuracy in non-iterated and short repetitive sequences, where a misinsertion occurs only once per ~500,000 nucleotides synthesized [69]. However, its fidelity decreases in homopolymeric tracts exceeding four nucleotides in length, resulting in an increased risk of frameshift errors. This suggests that such regions in mtDNA are particularly susceptible to replication errors by pol γ. The PolG2 accessory subunit, which enhances pol γ processivity, paradoxically reduces both frameshift and base substitution fidelity by facilitating extension of mismatched primer termini [69].

Multiple lines of evidence suggest that the primary driver of mitochondrial DNA mutagenesis is spontaneous errors of DNA replication [70,71], most likely due to the absence of mismatch repair in mammalian mitochondria [72]. Analysis of age-specific mtDNA sequences reveals mutation signatures more consistent with polymerase errors as opposed to oxidative lesion-driven mutations [73,74]. Furthermore, mtDNA appears to be resistant to mutagenesis following exposure to exogenous DNA-damaging agents [75,76].

Biochemical studies have demonstrated that substituting Asp198 and Glu200 with alanine in the ExoI motif of the exonuclease domain of pol γ abolishes 3′→5′ exonuclease activity [77]. This proofreading function normally enhances replication fidelity by at least 20-fold [69]. Expression of exonuclease-deficient pol γ in human cells leads to accumulation of mtDNA point mutations [78]. In pancreatic islet cells, this defect impairs glucose tolerance, induces diabetes, and increases β-cell apoptosis [79,80]. In mice, cardiac-specific overexpression of exonuclease-deficient pol γ results in a greater than 23-fold increase in mtDNA point mutations, detectable mtDNA deletions, elevated apoptosis, and the development of cardiomyopathy [81,82,83].

Previous studies have associated mtDNA mutation accumulation with aging. The strongest evidence linking mtDNA mutations to aging came from “mutator mice,” generated independently by two groups, which carried homozygous mutations disrupting the exonuclease function of pol γ [84,85]. These mice exhibited premature aging between 6 and 9 months of age, including hair graying and loss, hearing impairment, spinal curvature, cardiomegaly, reduced body weight, and decreased bone density. Mutation frequency in these exonuclease-deficient mice has been quantified and demonstrates significantly elevated mutation rates in their mitochondrial genome [86,87]. Notably, asymptomatic heterozygotes exhibited a mutation frequency far higher than aged wild-type mice, indicating that elevated mutagenesis alone is insufficient to produce aging phenotypes [86]. In addition to point mutations, homozygous mutator mice show a 90-fold increase in mtDNA deletions compared with wild-type or heterozygous controls [88]. While wild-type deletions typically occurred at direct repeats ≥6 nucleotides, deletions in exonuclease-deficient mice arose independently of repeats, implicating them in aging pathology. More recently, mutator mice have also shown nuclear genomic instability, likely resulting from nucleotide pool imbalances and compensatory sequestration of nucleotides within mitochondria [89].

### 3.2. POLG-Related Diseases

Mutations in *POLG* represent the most common cause of inherited mitochondrial disease [90,91]. To date, more than 300 pathogenic mutations have been identified in the *POLG* gene (https://tools.niehs.nih.gov/polg/, accessed on 25 September 2025). These pathogenic *POLG* variants are now known to cause a broad and overlapping spectrum of clinical phenotypes—many of which were defined clinically before their genetic basis was understood. Most individuals exhibit a subset, rather than the full set, of features associated with a given phenotype. The age of onset varies widely, ranging from infancy to late adulthood. The clinical spectrum includes the following (reviewed in [92,93]):Alpers–Huttenlocher syndrome: A severe childhood-onset encephalopathy marked by intractable epilepsy and progressive liver failure.Childhood myocerebrohepatopathy spectrum: Presents during the first months to three years of life with developmental delay or regression, lactic acidosis, and myopathy. Additional features may include liver failure, renal tubular acidosis, pancreatitis, cyclic vomiting, and sensorineural hearing loss.Myoclonic epilepsy myopathy sensory ataxia (MEMSA): A group of disorders involving epilepsy, myopathy, and ataxia without ophthalmoplegia. This category includes what was previously described as spinocerebellar ataxia with epilepsy.Ataxia–neuropathy spectrum: Encompasses mitochondrial recessive ataxia syndrome (MIRAS) and sensory ataxia with neuropathy, dysarthria, and ophthalmoplegia.Autosomal recessive progressive external ophthalmoplegia (arPEO): Characterized by progressive weakness of the extraocular muscles, leading to ptosis and ophthalmoparesis. Although initially isolated to eye movement, many individuals later develop additional systemic symptoms of *POLG*-related disease.Autosomal dominant progressive external ophthalmoplegia (adPEO): Typically involves generalized myopathy along with varying degrees of sensorineural hearing loss, axonal neuropathy, ataxia, depression, Parkinsonism, hypogonadism, and cataracts.

Early-onset diseases associated with *POLG* mutations such as Alpers–Huttenlocher syndrome cause depletion of mtDNA, whereas the adult-onset diseases such as ataxia and PEO are associated with multiple mtDNA deletions. Symptoms from depletion usually do not occur until >50% of the mtDNA is depleted while adult-onset symptoms from multiple DNA deletions are associated with deletions in >60% of the mtDNA genomes [94].

Most *POLG* mutations are linked to autosomal recessive progressive external ophthalmoplegia (arPEO), ataxia, or Alpers syndrome, and are frequently observed as compound heterozygous recessive variants. Most patients with recessive *POLG* mutations carry at least one of the common mutations: A467T, W748S, G848S, or the T251I-P587L allelic pair. Results from biochemical studies show that the A467T PolG disease variant exhibits only ~4% of wild-type polymerase activity, with minimal impact on exonuclease function [95]. Moreover, the A467T PolG variant fails to interact with its accessory subunit, p55, which is essential for highly processive DNA synthesis [95]. Despite its functional defects, A467T PolG is present in approximately 0.6% of the Belgian population [96].

The W748S mutation in *POLG* reduces polymerase catalytic activity and severely impairs DNA binding [97]. It is usually found in *cis* with the E1143G *POLG* polymorphism, which partially compensates for its deleterious effects and appears to modulate disease severity [97]. The G848S PolG variant, which is the third most common *POLG* mutation, is located in the thumb subdomain of the polymerase active site and displays less than 1% of normal polymerase activity due to defective DNA binding [98]. The T251I-P587L PolG substitutions, typically occurring in *cis* and present in up to 1% of the Italian population, are frequently found in PEO [99]. Individually, these mutations reduce polymerase activity by approximately 30%; however, when combined, they synergistically impair function to about 5% of normal, through decreased enzyme stability, reduced DNA binding affinity, and diminished catalytic efficiency [100].

Dominant *POLG* mutations are associated with adult-onset PEO and are almost exclusively located in the polymerase domain. Two key substitutions, R943H and Y955C PolG, directly alter residues that interact with incoming deoxynucleotide triphosphates (dNTPs) [101]. These variants retain less than 1% of wild-type polymerase activity and exhibit dramatically reduced processivity. The resulting stalling of DNA synthesis and low catalytic activity likely underlie the severe clinical phenotypes in heterozygotes [101]. Additionally, Y955C PolG increases nucleotide misinsertion errors by 10- to 100-fold in the absence of exonucleolytic proofreading [102].

A mouse model harboring the human A467T *POLG* mutation recapitulated the accessory subunit binding defect observed in humans [103]. In this model, the isolated PolG catalytic subunit becomes a substrate for the mitochondrial Lon protease (LonP1), leading to removal of the unstable A467T polymerase. Cross-linking mass spectrometry mapped this interaction to the accessory interacting determinant (AID) domain, which mediates PolG2 accessory subunit binding [104]. Thus, association with PolG2 not only enhances processivity, DNA binding, and salt tolerance but also protects the enzyme from proteolysis, suggesting an important regulatory role in polymerase activity as well as a site for therapeutic intervention (see Section 8).

## 4. Accessory Subunit, PolG2

The accessory subunit of human pol γ is encoded by the *POLG2* gene on chromosome 17q23-24 and translates into a 485 amino-acid protein weighing 55 kDa [105]. PolG2 functions as a processivity factor for mtDNA replication catalyzed by PolG. As a processivity factor, PolG2 functions include increasing DNA binding of the holoenzyme, accelerating polymerization rate, and suppressing exonuclease activity [67]. The gene for the accessory subunit has been identified in humans as well as mice, *Drosophila*, *Xenopus*, and other eukaryotes. In contrast, yeast, fungi, and *C. elegans* lack an accessory subunit altogether, as their pol γ functions as a single catalytic subunit [62,106,107]. PolG2 is unique as a eukaryotic processivity factor because of its similarity in primary amino acid sequence and conservation of tertiary structure to prokaryotic aminoacyl-tRNA synthetases and, to a lesser extent, thioredoxin, the accessory subunit of T7 DNA polymerase [108,109,110]. Additionally, PolG2 has also been reported to serve as a primer recognition factor and clamp loader in mtDNA replication in *Drosophila* pol γ [108].

The PolG2 accessory subunit has been characterized as a homodimer in solution, comprising the proximal and distal subunits [67,109]. The p55 dimer forms a strong interface of approximately 4000 Å^2^ [111] and ultracentrifugation studies have confirmed its tight association, with a dissociation constant (K_D_) of less than 0.1 nM. Functional studies indicate that the proximal p55 subunit stabilizes DNA binding, whereas the distal subunit enhances nucleotide incorporation [67,111]. Each monomer is further divided into three functional domains: domain one lies downstream of the N-terminal mitochondrial targeting sequence (MTS) and functions in p55 dimerization and binding of PolG, domain two contains the four-helix bundle that functions in homodimerization of p55, and the C-terminal domain three is involved in binding PolG [62]. PolG2 has also recently been reported to possess dimeric and hexameric (trimer of PolG2 dimers (Figure 2)) DNA binding modes [112]. The dimeric PolG2 preferentially binds duplex DNA, while multimeric PolG2 binds to DNA crossings or forked DNA resembling DNA substrates found in the D-loop of mtDNA. These observations suggest that PolG2 DNA binding serves both PolG-dependent and -independent utility in mtDNA replication and maintenance [112,113]. Additionally, PolG2 also permits increased salt tolerance of the pol γ holoenzyme during DNA synthesis, protects the catalytic subunit from inhibition by N-ethylmaleimide, and from degradation by LonP1 [103,104,105].

The mitochondrial disorders associated with *POLG2* mutations involve amino acid residues distributed across the three domains of the protein. Generally, disease mutations in *POLG2* give rise to decreased interaction with the catalytic subunit, PolG, as well as instability of the homodimeric enzyme. The first pathogenic *POLG2* mutation (c.1352G > A, p.G451E) was observed in a 60-year-old patient with late-onset adPEO, with multiple mtDNA deletions in the skeletal muscle reflected in mild weakness of facial and limb muscles, and ptosis [47]. In the PolG2 structure, this heterozygous mutation is located in a loop region in domain three that is not involved in the dimerization of the protein. Biochemical analysis of the G451E PolG2 variant revealed that this variant retained DNA binding capability but exhibits impaired interaction with the catalytic subunit, PolG, hence impairing processivity of the holoenzyme. The second heterozygous *POLG2* disease mutation (c.1207_1208ins24) was identified in a patient with similar phenotypic manifestations as the first, including adPEO, late onset ptosis, and mild myopathy. This 24 bp insertion causes mis-splicing and skipping of exon 7 in cDNA analysis. PolG2 with exon 7 that has been spliced is predicted to cause disruption of the C-terminal region required for processivity [114]. This mutation was also reported with the dominant clinical symptom of camptocormia in a 68-year-old patient, further supporting the point that mitochondrial disorders may be similar but present with varied phenotypes [115].

In an analysis of 112 patients with mitochondrial diseases and no *POLG* mutations, eight variants were observed, seven of which were novel heterozygous PolG2 disease variants [116]. However, only the R369G PolG2 disease variant is linked to adPEO, similar to G415E PolG2. The R369G PolG2 variant exhibits less binding affinity for the catalytic subunit PolG; however, the G451E variant appears to be more defective than the R369G PolG2 homodimeric variant in DNA processivity assays [117]. Further analyses of heterozygous PolG2 disease mutants were performed due to the hypothesis that affected patients may harbor mixtures of variants and wild-type (WT) p55 molecules. Heterodimeric PolG2 disease variants were purified with a tandem affinity strategy, and biochemical analysis showed that the G451E PolG2 variant was a dominant negative that caused severe inhibition of DNA processivity with WT p140 in vitro [118]. The inhibition by WT/G451E p55 heterodimer is distinct from the inability of the G451E p55 homodimer to interact with WT p140, suggesting that a single mutant monomer in the heterodimer is dominant negative and disrupts the interaction with the catalytic subunit. The results from the WT/G451E p55 heterodimer were in contrast to the R369G p55 heterodimeric variant, which displayed wild-type-like activity. Additionally, in vivo studies with R369G and G451E PolG2 heterodimeric variants revealed that they failed to localize to mtDNA-containing nucleoids and that their expression is associated with diminished reserve respiratory capacity [118].

The first homozygous *POLG2* disease mutation (c.544C > T, p.R182W) was identified in a 3-month-old patient with fulminant hepatic failure due to severe mtDNA depletion [119]. This pathogenic variant is classified within MDS, which is an autosomal recessive disorder characterized by significantly decreased mtDNA content that impairs energy production in multiple organ systems. The Arg182 residue is conserved across vertebrates and is located at the base of the four-helix bundle that is critical for dimer formation, suggesting that the mutation to tryptophan leads to loss of electrostatic interactions that may disrupt dimerization. HEK293 cells expressing R182W PolG2 showed significantly impaired respiratory capacity, while patient fibroblasts showed a reduction in mtDNA copy number. Interestingly, compared to WT PolG2, the R182W variant exhibits no difference in DNA binding affinity, p140 binding affinity, steady-state stimulation of p140 DNA polymerase activity, and stimulation of processive DNA synthesis by pol γ [120]. However, protein purification and results from size-exclusion chromatography–multi-angle light scattering (SEC-MALS) revealed the R182W PolG2 variant to be potentially unstable, potentially disrupting dimerization. This finding was confirmed by the significant reduction in thermostability of R182W PolG2 compared to wild-type PolG2, and this instability further impacted its ability to stimulate PolG in thermostability activity assays [120].

The second reported homozygous *POLG2* disease mutant (c.694G > A, p.G232S) was documented in a patient with epileptic seizures without ophthalmoplegia. The p.G232S variant is also the second recessive *POLG2* mutation reported, though it is a single-nucleotide polymorphism (SNP) with allele frequency ≤0.0002 [121]. The first ultrarare homozygous missense *POLG2* mutation (p.Asp433Tyr) that resulted in MDS was identified in an adult patient with a childhood onset and complex neuro-ophthalmic phenotype [122]. Asp433 is located in the proximal monomer of PolG2, which is also close to the catalytic thumb subdomain of PolG. A network of hydrogen-bonding interactions was identified with this residue that enables further polar interactions with PolG in the holoenzyme. An Asp-to-Tyr mutation is predicted to disrupt the hydrogen bond interactions in the proximal monomer and to further abolish interactions with PolG. The A433T PolG2 variant exhibited reduced thermal stability, intermediate efficiency in stimulating processivity, and had no measurable change in exonuclease activity of the holoenzyme, which may be explained by the loss of interactions predicted with this variant [122]. Additional *POLG2* mutations have since been reported; however, only a subset has undergone clinical evaluation, and extensive biochemical characterization still needs to be performed [123,124,125,126,127,128].

In addition to its role as a processivity factor, the accessory subunit has been proposed to stabilize the catalytic subunit, PolG. This is evidenced by both the severe decrease in PolG expression in *Polg2* knockout cells and the ability of WT PolG2 to stabilize PolG from heat-induced inactivation [95,129,130]. The *Polg2* gene has also been reported to be critical for development in *Drosophila,* as mutations that impair the function of this gene caused lethality in the early pupal stage, concomitant with mtDNA depletion [131]. Results from studies where heterozygous (*Polg2^+/−^*) and homozygous (*Polg2^−/−^*) *Polg2* knockout mice were generated to investigate the role of the accessory subunit in mammalian systems show that *Polg2^+/−^* mice are haplosufficient and exhibit normal development after two years. However, homozygous *Polg2* knockout mice exhibit embryonic lethality at 8–8.5 dpc, alongside loss of mtDNA and mtDNA gene products [65]. Furthermore, knockdown of *Polg2* in porcine oocytes showed a decrease in mtDNA copy number, which led to a reduction in ATP synthesis, oocyte maturation rate, and maturation-promoting factor activity, resulting in an aging-type phenotype [132]. PolG2 has also been reported to be critical for nucleoid maintenance, and its role in the initiation of mitochondrial DNA replication has been proposed to occur through the recruitment of proteins to the D-loop [133]. Collectively, these results demonstrate the fundamental role of PolG2 in mammalian embryogenesis, mtDNA maintenance, replication, and initiation.

## 5. The Human Mitochondrial Single-Stranded DNA-Binding Protein, mtSSB

Single-stranded DNA-binding proteins (SSBs) are among the most abundant and evolutionarily conserved cellular proteins found in all domains of life. A major characteristic of this group of proteins is their high nonspecific affinity for ssDNA [134,135,136]. SSB proteins play essential roles in various cellular processes, including DNA replication, repair, and recombination. Single-stranded DNA molecules are known to form secondary structures and can reanneal to themselves, which in both cases may interfere with these aforementioned cellular processes [135,136,137]. In addition, ssDNA is vulnerable to being cleaved by intracellular nucleases. Therefore, SSB proteins are required to protect and stabilize ssDNA from nuclease degradation and to keep ssDNA in a functionally active state. Several enzymes and protein factors can interact with ssDNA during different stages of DNA replication, repair, and recombination [137]. SSB proteins are known to interact both functionally and physically with genome maintenance proteins, by directing them to their respective sites of action or by stimulating their biochemical functions [137,138,139,140,141]. SSB is present in both the eukaryotic mitochondria and in the nucleus; however, the human mitochondrial SSB (mtSSB) is structurally and evolutionarily distinct from its nuclear counterpart, replication protein A (RP-A) [137].

MtSSB is essential for mtDNA replication, initiation, and elongation, as it coats the displaced parental H-strand during mtDNA replication. Furthermore, during mtDNA replication, mtSSB prevents primer synthesis by POLRMT on the displaced H-strand and controls the initiation of L-strand mtDNA synthesis from its designated origin of replication, O_L_ [24,142,143,144]. Unlike POLRMT, Twinkle, and pol γ which are evolutionarily related to their bacteriophage T7 counterparts, mtSSB does not share bacteriophage ancestry but instead exhibits structural and biochemical similarities to *Escherichia coli* SSB [145]. The gene encoding human mtSSB, *SSBP1*, was first cloned in 1993 [146]. The cDNA of *SSBP1* is translated as a 148 amino-acid polypeptide, in which the first 16 amino acids make up the MTS required for import of this protein into the mitochondria [146]. Human mtSSB binds ssDNA as a tetramer, composed of four identical 16 kDa subunits, each containing an oligosaccharide/oligonucleotide binding fold (OB-fold) [147,148]. Similar to *E. coli* SSB, human mtSSB can bind ssDNA in two distinct modes, termed (SSB)_30_ and (SSB)_60_, where the number represents how many nucleotides are occluded per tetramer [136,149,150,151,152]. In a previous study by our group, atomic force microscopy of the human mtSSB with ssDNA demonstrated a single wrap of ssDNA around the mtSSB tetramer at physiological salt conditions [153]. Despite having only 32% amino acid identity, the overall tetramer folding of the human mtSSB resembles the tetrameric structure of *E. coli* SSB. However, *E. coli* SSB contains an additional 60-amino-acid acidic C-terminal tail required for protein–protein interactions, which is lacking in the human mtSSB [148,154,155].

Despite the lack of evidence for physical protein–protein interactions between mtSSB and other replisome proteins, mtSSB has been reported to functionally interact with pol γ and the Twinkle helicase at the replication fork [143,156]. Moreover, mtSSB facilitates the binding of pol γ to the primer-template DNA and enhances the processivity of pol γ during mtDNA initiation and elongation, thereby having an overall stimulatory effect on mtDNA synthesis [143]. In addition, mtSSB has been reported to stimulate the unwinding activity of the mitochondrial Twinkle helicase at the replication fork. Since ssDNA can re-anneal to itself and hinder mtDNA replication, mtSSB prevents ssDNA reannealing through binding of ssDNA, thereby stimulating the unwinding activity of Twinkle [144,156]. The functional interaction between human mtSSB and the mitochondrial Twinkle helicase appears to be highly specific, since the structurally similar *E. coli* SSB could not stimulate the unwinding activity of the human mitochondrial Twinkle helicase [157]. Functional interactions of mtSSB with pol γ and Twinkle have also been demonstrated through in vitro studies, in which, in the absence of mtSSB, pol γ and Twinkle can synthesize ssDNA products only up to ~2 kb. The addition of mtSSB to this reaction enhances the processivity of the minimal replication machinery, resulting in DNA products longer than ~16 kb in vitro [158]. Although biochemical analyses provide evidence that mtSSB functionally interacts with pol γ and Twinkle, direct physical protein–protein interactions are currently elusive [158].

While SSB proteins lack direct catalytic activity, they are essential for DNA metabolism [137]. In addition to humans, mtSSB proteins are present in higher eukaryotes, including yeast, mice, *Xenopus laevis*, and *Drosophila* [159]. In yeast, deletion of the *RIM1* gene, encoding mtSSB, results in complete loss of mtDNA and yields strains that are unable to grow on non-fermentable carbon sources [48]. Similarly, RNAi knockdown of *Drosophila* mtSSB significantly reduces its expression in Schneider cells, resulting in growth defects and severe mtDNA depletion [160]. Knockdown of mtSSB in HeLa cells results in a moderate decrease in mtDNA synthesis, a gradual reduction in mtDNA copy number, and a significant reduction in the synthesis of 7S DNA [161]. Knockdown of mtSSB in osteosarcoma cells has also been shown to significantly affect the synthesis of 7S DNA, reinforcing the role of mtSSB as both an mtDNA maintenance factor and a key player in mtDNA replication [161].

Since 2018, over 10 mutations have been identified in the *SSBP1* gene that have been associated with mitochondrial diseases. Patients with *SSBP1* mutations typically suffer from optic atrophy and have also been reported to experience neurological dysfunction mainly affecting the visual system (retinopathy, foveopathy, ptosis, ophthalmoplegia) and auditory system (sensorineural hearing loss) [46,162,163,164,165]. With the exception of the I132V disease-associated mutation in *SSBP1*, all other reported mutations are heterozygous in patients [159,164]. These mutations are mostly missense mutations, with the lone exception of c.3G > A, which abolishes the start codon of *SSBP1,* resulting in decreased protein translation. The c.3G > A mutation in *SSBP1* observed in a patient also co-segregated with a point mutation in the mtDNA (m.1555A > G), which resulted in decreased levels of 7S DNA in fibroblasts, mtDNA depletion, and multiple mtDNA deletions [162]. In 2019, a single de novo *SSBP1* mutation (E27K) was observed in a 14-year-old patient who had mitochondrial disease manifestations across the full Pearson, Kearns–Sayre, and Leigh syndromes spectrum. In addition, the patient presented with infantile anemia, bone marrow and growth failure, ptosis, ophthalmoplegia, metabolic strokes, and chronic kidney disease [46]. Results from mtDNA genome sequencing using next-generation sequencing revealed a single large-scale mtDNA deletion (SLSMD) of 5440 base pairs (m.8629_14068del5440). While E27K mtSSB remains a stable tetramer in solution and exhibits similar binding affinity for ssDNA as the wild-type protein, this mutation resulted in a slight decrease in thermal stability [46]. Furthermore, molecular modeling of E27K mtSSB comprising different combinations of mutant subunits predicted changes in the surface-accessible charges, strength of inter-subunit interactions, and overall protein dynamics [46]. Altogether, it was proposed that these observed alterations could interfere with mtDNA replication, explaining the resulting SLSMD observed in the patient [46,159].

Like several other disease-associated residues, Glu27 is evolutionarily conserved across higher eukaryotes [46,159]. Within the homotetrameric structure of mtSSB, Glu27 is located at the monomer–monomer interface at the edge of a positively charged surface patch containing Arg38 and Arg107, which are both residues with mitochondrial disease alleles (R38Q and R107Q). In the structure, Glu27 participates in an inter-subunit interaction with Arg38 and Arg107, which has been proposed to be structurally significant for tetramer formation [46,159,165]. This inter-subunit interaction is facilitated by the guanidinium nitrogens of the Arg38 and Arg107 side chains, which interact with the side-chain carboxylate group of Glu27 in the neighboring subunit [46,159,165]. Even though the first X-ray crystallography structure of apo mtSSB was determined in 1997, a ssDNA-bound mtSSB structure was elusive for more than 20 years [148]. In 2024, our group was the first to determine a high-resolution X-ray crystallography structure of mtSSB bound to a dT_20_ ssDNA (Figure 2) [147]. This ssDNA-bound mtSSB structure revealed that three residues (Arg38, Gly40, and Gln62) with mitochondrial disease alleles (R38Q, G40V, and N62D) reside within 3.5Å of the ssDNA chain. These three residues were shown to participate in interactions with either the ssDNA backbone or the thymine nucleotides [147].

The autosomal dominant mutations, R38Q and R107Q, in *SSBP1*, were first identified in five unrelated families, all of whom exhibited optic atrophy. In addition to optic atrophy, nearly half of the affected individuals also presented with foveopathy [164,165]. Biochemical analyses using patient fibroblasts revealed that the R38Q and R107Q mtSSB variants impair tetramer stability, perturb mtDNA replication, and lead to mtDNA depletion. These results also revealed reduced levels of TFAM and PolG in patient fibroblasts [164,165]. Autosomal dominant mutations in the nuclear-encoded replication genes impair mtDNA replication by inducing stalled replication forks [166,167]. Therefore, frequent pausing of mtDNA replication would explain the depletion of mtDNA observed in patient fibroblasts [165]. In addition to mtDNA depletion, patients carrying the R107Q *SSBP1* mutation exhibited a significant reduction in Complex I and IV, resulting in severe respiratory deficiency [164]. Several other mtSSB disease variants (G40V, N62D, R107Q, E111Q, and I132V mtSSB) have been evaluated based on their ability to stimulate DNA synthesis. Although all these disease variants were able to support full-length DNA synthesis, they were consistently less efficient than the wild-type protein [164]. In the same study, zebrafish models were employed to assess the relevance of *ssbp1* in relation to the optic atrophy phenotype observed in patients with *SSBP1* mutations. Loss of *ssbp1* in zebrafish affected the development of the optic nerve and resulted in signs of nephropathy. While all dominant mtSSB variants were ineffective in rescuing the optic nerve phenotype, the recessive I132V mtSSB disease variant behaved as a hypomorph [164]. Finally, results from another study evaluating the ability of mtSSB disease variants to support replication initiation and elongation revealed that the G40V, N62D, R107Q, and E111Q mtSSB disease variants cannot support replication initiation in vitro. However, these variants did not impair replication elongation [168].

## 6. The Mitochondrial DNA Helicase, Twinkle

Helicases are a class of molecular motor proteins that employ the energy derived from nucleotide hydrolysis to drive the unwinding of double-stranded DNA/RNA [169]. They play diverse roles in DNA replication, repair, recombination, and protein synthesis. Helicases are classified into six families (SF 1–6) based on conserved sequence and structural motifs [170]. Although multiple helicases have been identified in mitochondria, Twinkle is the only helicase in mtDNA replication, where it functions in unwinding duplex DNA and during pol γ–catalyzed mtDNA synthesis [171,172]. Twinkle was first reported in 2001; it belongs to superfamily 4 (SF4) DNA helicases and shares 46% amino acid sequence similarity and 15% sequence identity with the bacteriophage T7 gene protein 4 (gp4) primase–helicase fusion protein [173,174]. In mammalian cells, Twinkle is encoded by the *TWNK* gene (also commonly referred to as *C10orf2* or *PEO1*), and is translated into a 684-amino-acid protein with a molecular weight of approximately 77 kDa [144]. Twinkle helicase is highly conserved in many eukaryotes including *Drosophila* and mice but lacks orthologs in yeast [175,176].

The replicative Twinkle helicase is structurally organized into an N-terminal region (NTR), which includes the primase-like domain, connected by a flexible linker region to the C-terminal helicase domain. Although Twinkle shares significant homology with the bacteriophage T7 primase–helicase (gp4) through its C-terminal helicase domain (CTD), the NTR and primase-like domain lack critical conserved cysteine residues required for formation of the zinc finger binding domain (ZBD), which is essential for primase activity [173]. The different domains have unique functions: the N-terminal domain contributes to ssDNA binding, helicase activity, and processivity; the linker region is critical for multimerization; and the C-terminal helicase domain functions in oligomer stability [171,177]. Disease-causing mutations in the *TWNK* gene have been mapped mostly to the linker region and C-terminal domain [178]. Results from previous studies have reported the ability of Twinkle to interact with a variety of DNA substrates including circular and linear ssDNA, but it shows a greater affinity for linear dsDNA [144,179,180]. However, for the initiation of unwinding, Twinkle requires an open fork-like dsDNA comprising a single-stranded 5′ DNA loading site and a short 3′ tail [144].

Previous structural and biochemical studies have predicted that Twinkle exists in multiple oligomeric states and undergoes oligomeric shifts [156]. These oligomeric transitions have been observed to be regulated by salt concentrations and the presence of Mg^2+^ and dNTPs [181,182]. As a ring-structured 5′→3′ DNA helicase, Twinkle binds dNTPs at the interface and DNA in the central channel, and the mechanism by which Twinkle loads onto DNA has been predicted to occur through oligomeric transitions typical for gp4 helicases [144,183]. Recent cryo-EM structures of the Twinkle W315L disease variant revealed two oligomeric states, heptamer and octamer (Figure 2) [184]. Additionally, results from SEC-MALS revealed that the W315L Twinkle heptamer was observed upon DNA binding, which supports the ejection theory of a monomer subunit from the ring-shaped octamer [184]. Twinkle’s unwinding activity has also been shown to facilitate the degradation of linear mtDNA by endonucleases specific for ssDNA, including the 3′→5′ exonuclease activity of PolG [185]. In addition to its helicase activity, Twinkle also functions in strand exchange, reannealing, and branch migration, which are proposed to be important for DNA recombination [186,187,188,189]. Given that Twinkle functionally interacts with mtSSB and pol γ, it has been proposed that even subtle changes in Twinkle’s activity may disrupt the coordination with mtSSB and pol γ at the replication fork [143,156,158,190,191,192].

The mitochondrial DNA helicase, Twinkle, colocalizes with mtDNA in mitochondrial nucleoids [13,173,193]. The importance of Twinkle in mtDNA maintenance has been demonstrated through multiple lines of experimental evidence. In particular, studies using in vivo mouse models have provided evidence that Twinkle is critical for embryonic development, as homozygous knockout of Twinkle results in embryonic lethality at 8.5 dpc and co-segregates with mtDNA depletion, impaired mtDNA expression, and respiratory chain deficiency [178]. In contrast, mice harboring a conditional tissue-specific knockout of Twinkle in their heart and skeletal muscle remain viable [178]. Conversely, systemic overexpression of Twinkle in a transgenic mouse results in a significant increase in mtDNA copy number; however, these animals exhibited enlarged nucleoid structures, impaired mtDNA replication, and disrupted transcription [20,194]. In *Drosophila*, depletion of Twinkle in Schneider cells results in slow growth, decreased viability, and a fivefold reduction in mtDNA copy number [195]. Collectively, these results confirm that Twinkle cannot be substituted by other mitochondrial helicases and underscore the importance of Twinkle as a key player in mtDNA replication and maintenance [171].

The *TWNK* gene was initially identified through sequencing analysis in a group of patients suffering from PEO, which co-segregated with multiple mtDNA deletions and 11 different mutations across 12 families [173]. Typically, an autosomal dominant mode of inheritance is revealed by the combined existence of disease and mutant allele, in which one mutant allele is sufficient to develop adult-onset PEO. In contrast to their heterozygous relatives, homozygous individuals exhibit a more severe phenotype with earlier onset of PEO [173]. In addition to PEO, the T457I, Y508C, and A318T Twinkle disease variants are associated with neurological disorders, hepatocerebral MDS, and infantile-onset spinocerebellar ataxia (IOSCA), which manifest in the initial stages of life [196,197]. Individuals homozygous for a *TWNK* mutation causing a dual T457I substitution presented with hepatocerebral MDS and died by the age of three years; however, their heterozygous parents and grandparents remain unaffected [198]. A similar result was also observed for the Y508C Twinkle variant in which homozygous individuals developed IOSCA, whereas heterozygotes were unaffected [199]. Furthermore, biochemical analyses of recombinant Y508C Twinkle revealed that this variant has similar helicase activity as the wild-type; however, a modest decrease in ssDNA binding affinity has been reported [197,200]. Finally, the A318T Twinkle variant has only been observed as a heterozygous mutation with Y508C. Individuals harboring both the A318T and Y508C *TWNK* mutations presented with a severe early-onset encephalopathy and mtDNA depletion in the liver. Taken together, these findings suggest that one T457I-, Y508C-, or A318T-substituted copy of *TWNK* is not sufficient to induce disease [196,201]. These findings also emphasize that one wild-type *TWNK* allele can rescue any biochemical defects caused by these mutant alleles [200].

Many of the *TWNK* mutations found in adPEO patients are located within or in close proximity to the linker region [202]. The cryo-EM structure shows that these mutations cluster at subunit interfaces, and substitution of these residues disrupts this subunit–subunit interface [184]. The importance of the linker region was also demonstrated through in vitro biochemical analyses of several Twinkle disease variants. Unlike the wild-type enzyme, I367T and R374Q Twinkle disease variants eluted as monomers in gel filtration assays and displayed no ATPase and helicase activity [174]. Furthermore, the I367T, S369P, R374Q, and L381P Twinkle disease variants were unable to perform rolling-circle DNA synthesis. This inability to facilitate DNA synthesis was attributed to impaired ATPase and helicase activities as well as the failure of certain variants to form hexamers in solution [174]. In contrast, a previous study from our group demonstrated that 20 Twinkle disease variants have similar helicase activity as the wild-type enzyme under optimized in vitro conditions [200]. Despite preserved helicase activity, several of these 20 Twinkle disease variants displayed reduced DNA binding affinity, diminished nucleotide hydrolysis rates, and decreased thermal stability compared to the wild-type enzyme [200]. Given the moderate biochemical defects observed in vitro, we proposed that these results are consistent with the delayed onset of adPEO due to *TWNK* mutations [200]. It is noteworthy that the current literature contains significant discrepancies regarding different physical and biochemical properties of wild-type and mutant Twinkle. These discrepancies may be explained by the specific methods and conditions utilized for biochemical evaluation [166,174,177,200].

In addition to in vitro biochemical analyses, the use of in vivo models has been useful in elucidating disease outcomes associated with Twinkle variants. Transgenic mice overexpressing either the dominant A360T *Twinkle* mutation or a 13-amino-acid duplication in the linker region were found to accumulate multiple mtDNA deletions [203]. The transgenic mouse containing the duplication is also referred to as the deletor mouse in the literature and exhibits respiratory dysfunction and mitochondrial myopathy starting at one year of age [203]. The phenotypes observed with the deletor mouse closely resembled those seen in PEO patients [203,204]. A significant hallmark of PEO disease is the accumulation of multiple large-scale mtDNA deletions in postmitotic tissues of the patients, with the level of mutant mtDNA reaching up to 60% in the brain region and 40% in the skeletal muscle [205,206]. Furthermore, overexpression of several adPEO Twinkle variants in HEK293 cells resulted in the accumulation of replication intermediates. Expression of these adPEO Twinkle variants also resulted in decreased mtDNA copy number, enlarged nucleoids, and was proposed to induce replication stalling [166,167]. Beyond PEO, defects in Twinkle have also been linked to neurodegenerative diseases [207]. Analysis of neurons from the deletor mouse revealed cytochrome c oxidase deficiency, which was attributed to mtDNA depletion and the accumulation of mtDNA deletions in specific neuronal populations [203]. Collectively, these findings underscore the critical role of Twinkle in mtDNA replication and maintenance.

## 7. Other Proteins Involved in mtDNA Maintenance

Several proteins are involved in maintaining mtDNA genomic stability and mitochondrial function. Beyond the core replication machinery discussed in this review, defects in proteins involved in mtDNA transcription, replication, nucleotide pool maintenance, and DNA repair pathways of the mitochondrial genome can all influence mtDNA integrity (Table 1). Additionally, disease alleles in proteins involved in mitochondrial dynamics and mitochondrial quality control pathways may influence mtDNA integrity, as these pathways are involved in removing damaged mitochondrial genomes.

In mitochondria, RNase H1 is crucial to mtDNA replication [208] and has been implicated in both primer removal during mtDNA replication [209,210] and primer maturation [211]. Additionally, RNase H1 is critical for the segregation of daughter mtDNA molecules following replication [212], R-loop resolution [213], regulating transcription of 7S RNA [214], and pre-rRNA processing [215]. Disease-associated alleles in *RNASEH1* have been identified and affected patients display classic mitochondrial disease phenotypes, including mitochondrial myopathy, PEO, cerebellar ataxia, muscle weakness, dysphagia, mtDNA depletion, and the accumulation of mtDNA deletions [216,217,218,219,220].

DNA replication helicase/nuclease 2 (DNA2) localizes to both the mitochondria and the nucleus. DNA2 is essential to mtDNA replication and the long-patch base excision repair pathway in the mitochondria [221,222,223,224]. Numerous pathogenic variants in *DNA2* have been identified with clinical outcomes such as mtDNA instability, mitochondrial myopathy [225,226,227,228], congenital ptosis, and hypotonia [229]. Other disease outcomes include microcephalic primordial dwarfism [230,231], mtDNA depletion syndrome associated with hearing loss and muscle weakness [232], and mesial temporal lobe epilepsy [233]. Recently, a patient carrying a *DNA2* mutation presented with adult-onset mitochondrial encephalomyopathy, mtDNA depletion, and mtDNA deletions [234].

Mitochondrial genome maintenance exonuclease 1 (MGME1) is a mitochondrial deoxyribonuclease that can cleave ssDNA in both 5′→3′ and 3′→5′ directions, dsDNA flaps, and RNA-DNA flaps [235,236]. MGME1 is crucial to mtDNA maintenance, the regulation of 7S DNA, and is involved in the degradation of linear mtDNA [185]. Loss-of-function mutations in *MGME1* are associated with mtDNA depletion and mtDNA rearrangements [235,237,238]. Clinical manifestations linked to *MGME1* mutations include PEO, skeletal muscle weakness, respiratory distress, and neurological symptoms [235,238].

Top3α localizes to both the nucleus and the mitochondria. The mitochondrial isoform of Top3α is essential for genome segregation during mtDNA replication [32] and for regulating topology of mtDNA at the replication fork [239]. Pathogenic variants in *TOP3A* have been associated with Bloom syndrome-like disorders [240,241] as well as primary mitochondrial disease characterized by adult-onset PEO [32,33,242,243].

TFAM, which facilitates mtDNA compaction, initiates transcription, plays a role in the regulation of mtDNA replication, and participates in DNA repair processes, has very few disease alleles. Although there are numerous documented allelic variants of the *TFAM* gene, only two pathogenic variants have been identified [244]. A c.533C > T (p.Pro178Leu) mutation was associated with mtDNA depletion and neonatal liver failure [245]. The P178L TFAM variant has been reported to have a deficiency in transcription initiation [246]. A c.694C > T (p.Arg232Cys) mutation has also been reported and is associated with primary ovarian insufficiency in females, abnormal sex hormone levels in males, neurological symptoms, and mtDNA depletion [247]. Among the non-disease-causing allelic variants of *TFAM* that have been reported, evidence suggests that some variants are implicated in the risk of Alzheimer’s disease and Parkinson’s disease [248,249], reduced survival rates in patients with head and neck cancers [250], and are co-occurring with colorectal cell line and tumor cancers [251].

The mitochondrial RNA polymerase initiates mtDNA replication via the synthesis of RNA primers for pol γ to begin replication. Recently, 14 pathogenic variants of POLRMT associated with mitochondrial disease have been reported, with phenotypes including adult-onset PEO, motor delays, and neurological symptoms [252,253]. Although all patient mutations in *POLRMT* were associated with reduced in vitro primer synthesis, patient fibroblasts harboring the *POLRMT* mutations did not exhibit mtDNA depletion or deletions. Notably, these fibroblasts exhibited a significant loss of OXPHOS mRNA transcripts [252].

While we have briefly described several disease-associated variants in proteins that are crucial to maintaining mitochondrial genomic integrity and supporting mtDNA replication, this list is not exhaustive. Additional genes not covered in this review include those involved in DNA repair processes of the mitochondria, nucleotide pool maintenance, mitophagy, mitochondrial quality control mechanisms, and those for which disease variants cause mtDNA depletion without a known mechanism (Table 1).

## 8. Therapies for Mitochondrial Diseases

Current therapy for PolG-related disorders is primarily symptomatic and supportive, with a focus on managing symptoms and preventing progression. Seizure management needs to avoid the use of valproate, which can cause fatal liver failure in patients with *POLG* mutations. Mitochondrial-cocktail supplements including coenzyme A, riboflavin, L-carnitine, creatine, arginine, and other cofactors, as well as antioxidants, are frequently used to improve mitochondrial function and enhance energy metabolism in patients. Below, we summarize recent research-supported therapies that may offer promising clinical benefits.

*Nucleotide therapy*: MtDNA replication is dependent on dNTP pools in the mitochondria, and one of the main sources of dNTPs is the salvage pathway involving phosphorylation of pyrimidines by thymidine kinase 2 (TK2) and purines by deoxyguanosine kinase (DGUOK). Mutations in the genes *TK2* and *DGUOK* result in early childhood mitochondrial DNA depletion (reviewed in [51]). Over the past decade, nucleotide therapy has proven to be highly effective against TK2 deficiencies as a result of *TK2* gene mutations [254,255]. Nucleotide therapy application for PolG defects has also been considered. Given that mitochondrial deoxynucleotide pools are usually limiting for mtDNA replication, increasing dNTP availability has shown promise in fibroblasts from patients harboring either *POLG* or *TWNK* mutations in ethidium bromide-forced depletion and recovery [256,257]. This approach has been extended to iPSC-derived neural stem cells with *POLG* mutations and has been shown to improve mtDNA depletion after ethidium bromide treatment [258]. Furthermore, in a 6-month, open-label trial, supplementation with deoxycytidine and deoxythymidine in children with PolG-related disorders demonstrated improved energy levels, enhanced EEG profiles, and favorable changes in multiple biomarkers of mitochondrial dysfunction [259,260].

*Metformin and nicotinamide riboside*: Mitophagy is a key process for the removal of damaged mitochondria that can result from a buildup of mtDNA mutations derived from PolG defects. Investigation into modulating mitophagy in *POLG*-mutant astrocytes has identified that combined treatment with nicotinamide riboside (NR) and metformin restored mitophagy and mitochondrial function [261]. These findings indicate that impaired mitophagy contributes to mitochondrial dysfunction in *POLG* mutations and highlight NR and metformin as potential therapies for PolG-related mitochondrial diseases.

*Targeted drug screening*: Recently, a small effector molecule, PZL-A, was identified in a high-throughput screen of 270,000 compounds based on its ability to stimulate pol γ activity and mitigate the phenotypes associated with impaired pol γ function [262]. The PZL-A molecule binds allosterically at the interface between the catalytic PolG and accessory PolG2 subunits. PZL-A effectively restores near wild-type activity to the most common mutant forms of PolG in vitro, reviving mtDNA synthesis. Furthermore, in cells from pediatric patients with lethal diseases associated with *POLG* mutations, PZL-A enhances mitochondrial DNA replication, boosts the biogenesis of oxidative phosphorylation complexes, and improves cellular respiration. Structural studies support these findings, including several high-resolution cryo-EM structures demonstrating that both wild-type and mutant PolG complexes bound to PZL-A, illuminating how the molecule interacts with and activates the enzyme [262]. This compound represents a promising therapeutic candidate for conditions caused by *POLG* mutations.

## 9. Concluding Remarks

One of the most important functions of the mitochondria is the production of cellular ATP via OXPHOS, which is subsequently used for various metabolic processes. Cellular ATP is generated through a series of electron transfer reactions requiring genes expressed by both the nuclear and mitochondrial genomes. Defects in this energy production pathway cause a broad spectrum of diseases, collectively known as mitochondrial diseases. Mutations in the nuclear-encoded mtDNA replication machinery continue to be one of the main causes of mitochondrial diseases. These mutations impair the rate and efficiency of the replisome proteins during progression of the replication fork, mtDNA synthesis, protein–DNA affinity, as well as protein and complex stability. Collectively, alterations in the biochemical functions of the replisome proteins may result in stalled replication forks, uncoupling of the replication machinery, DNA strand breaks, and exposure of ssDNA [263]. Consequently, stalled replication forks and DNA strand breaks can contribute to an increase in the number of point mutations, the formation of large-scale mtDNA deletions, and mtDNA depletion. Therefore, understanding the molecular mechanisms pertaining to mtDNA replication is critical to understanding the pathogenesis of mitochondrial diseases. Mechanistic knowledge of replisome proteins, combined with the use of animal models and biochemical characterization in the investigation of disease variants, provides a foundation for developing effective therapeutic interventions for mitochondrial diseases.

## Figures and Tables

**Figure 1 ijms-26-10275-f001:**
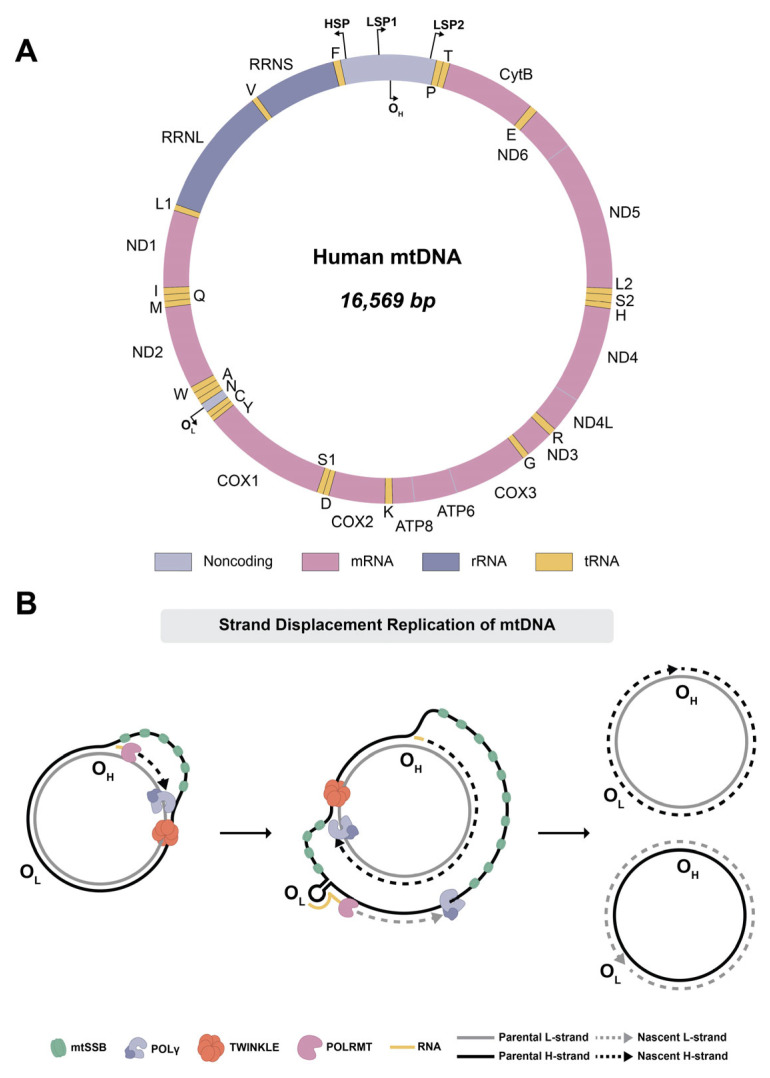
Human mitochondrial DNA. (**A**) Schematic of the human mitochondrial genome. Human mtDNA is circular and 16,569 base pairs in length. Human mtDNA encodes 13 mRNAs, all of which are subunits of electron transport chain complexes, as well as 2 rRNAs, and 22 tRNAs. The noncoding region of the genome contains the heavy-strand origin of replication (O_H_), the heavy-strand promoter (HSP), and two light-strand promoters (LSP1 and LSP2). The light-strand origin of replication (O_L_) is located approximately two-thirds away from O_H_, near COX1 and ND2. Gene names annotated on the outside of the diagram are on the heavy strand, and gene names annotated on the inside of the diagram are on the light strand. (**B**) Overview of the strand displacement model of mtDNA replication. Replication is initiated first at O_H_, where the heavy strand becomes displaced and is stabilized by mtSSB. POLRMT synthesizes a short RNA primer, and Twinkle helicase unwinds the DNA as pol γ synthesizes a new heavy strand of DNA. Once the replication machinery reaches O_L_, a stem-loop structure is formed and POLRMT can initiate primer synthesis that allows replication of a new light strand to proceed in the opposite direction. Replication of mtDNA results in two identical daughter molecules of mtDNA.

**Figure 2 ijms-26-10275-f002:**
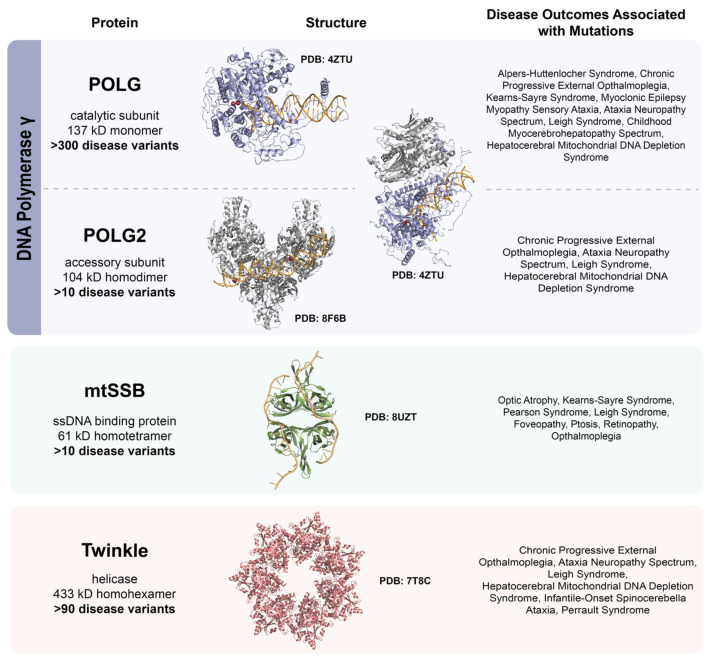
Structures of the minimal mitochondrial DNA replication machinery and the disease outcomes associated with mutations. The replicative polymerase, pol γ, is comprised of PolG, the catalytic subunit, and PolG2, the accessory subunit. The catalytic subunit alone is represented in PDB: 4ZTU bound to DNA. The accessory subunit bound to DNA as a trimer of dimers is represented in PDB: 8F6B. The holoenzyme in complex with DNA is shown in PDB: 4ZTU. Mitochondrial single-stranded binding protein bound to DNA is shown in PDB: 8UZT. The Twinkle helicase is represented in PDB: 7T8C.

**Table 1 ijms-26-10275-t001:** Mitochondrial disease genes associated with mtDNA maintenance.

Gene	Disorder	Locus	Function
mtDNA replication and repair			
*APTX*	ataxia	9p21.1	DNA repair
*DNA2*	mtDNA deletions, PEO, epilepsy	10q21.3	Mito/nuclear helicase–nuclease
*MGME1*	PEO, mtDNA depletion	20p11.23	Single-stranded DNA nuclease
*POLG*	PEO, Alpers, ataxia, epilepsy, mtDNA depletion	15q26.1	Pol γ catalytic subunit
*POLG2*	PEO, ataxia, mtDNA depletion	17q23.3	Pol γ accessory subunit
*POLRMT*	PEO	19p13.3	RNA polymerase
*RNASEH1*	PEO, encephalopathy, ataxia, mtDNA deletions	2p25.3	RNA/DNA hybrid endoribonuclease
*SSBP1*	Optic atrophy, mtDNA depletion/deletions	7q34	Single-stranded DNA-binding protein
*TFAM*	mtDNA depletion	10q21.1	DNA compaction, transcription factor
*TOP3A*	Bloom-syndrome-like disorders, PEO	17p11.2	Topoisomerase
*TWNK*	PEO, ataxia, mtDNA depletion	10q24.31	Replicative helicase
nucleotide pool metabolism and maintenance			
*ABAT*	mtDNA deletions, depletion	16p13.2	4-Aminobutyrate aminotransferase
*DGUOK*	mtDNA depletion, PEO	2p13.1	Deoxyguanosine kinase
*RRM2B*	PEO, mtDNA deletions, depletion	8q22.3	p53-Ribonucleotide reductase subunit
*SAMHD1*	mtDNA deletions	20q11.23	dNTP triphosphohydrolase
*SLC25A4*	PEO	4q34.1	Adenine nucleotide translocator
*SUCLA2*	mtDNA depletion	13q14.2	ATP-dep Succinate-CoA ligase
*SUCLG1*	mtDNA depletion	2p11.2	GTP-dep Succinate-CoA ligase
*TK2*	PEO, mtDNA depletion	16q21	Mitochondrial thymidine kinase
*TYMP*	MNGIE, mtDNA deletions/depletion	22q13.33	Thymidine phosphorylase
mitochondrial homeostasis and dynamics			
*AFG3L2*	Spinocerebellar ataxia, mtDNA deletions	18p11.21	Mitochondrial metalloprotease
*DNM1L*	Encephalopathy, neurological disorders, epilepsy	12p11.21	GTPase involved in mitochondrial fission
*FBXL4*	mtDNA depletion, encephalopathy	6q16.1–16.2	Mitochondrial LLR F-Box protein
*GDAP1*	CMT disease	8q21.11	Mitochondrial fission protein
*GFER*	mtDNA deletions, myopathy	16p13.3	Protein import to inner membrane
*MFF*	Encephalopathy, hypotonia, neurological disorders	2q36.3	Mitochondrial fission protein
*MFN2*	CMT disease, dominant optic atrophy, mtDNA deletions	1p36.22	Mitochondrial fusion protein
*MPV17*	mtDNA depletion, CMT disease	2p23.3	Unknown, inner membrane protein
*NME3*	Neurodegeneration, hypotonia	16p13.3	Nucleoside diphosphate kinase, fusion
*OPA1*	Dominant optic atrophy, mtDNA deletions, ataxia	3q29	Dynamin-related GTPase
*SLC25A46*	Leigh syndrome, optic atrophy, ataxia, CMT disease	5q22.1	Mitochondrial fission protein
*SPG7*	ataxia, spastic paraplegia	16q24.3	Mitochondrial metalloprotease
*STAT2*	Immunodeficiency	12q13.3	Mitochondrial fission protein
*TMEM65*	mtDNA depletion	8q24.13	Na^+^/Ca^2+^ exchange

## Data Availability

No new data were created or analyzed in this study. Data sharing is not applicable to this article.

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
