# Peer review of "Mitochondrial DNA Replication and Disease: A Historical Perspective on Molecular Insights and Therapeutic Advances"

_ijms, 2025, doi:10.3390/ijms262110275_

Round 1
Reviewer 1 Report
Comments and Suggestions for Authors
Somai and colleagues wrote an excellent review that focusses on the three main players of mitochondrial DNA replication: DNA polymerase gamma, the helicase Twinkle and mitochondrial single-stranded binding protein. The review is well-balanced, has been written with care, has sufficient depth, contains appropriate references and is up-to-date. I have only a few minor comments.
Page 8: “Symptoms from deletions usually don’t occur until >50% of the mtDNA is depleted while adult-onset symptoms from multiple DNA deletions are associated with deletions in >60% of the mtDNA genomes (91)”. Should this be: “Symptoms from depletion usually don’t occur until >50% of the mtDNA is depleted, while …”? Please check.
Please explain the abbreviation: “SEC-MALS” (Size-Exclusion Chromatography – Multi-Angle Laser Light Scattering).
Page 10: “R182W PolG2 expressed in HEK293 cells showed significantly impaired respiratory capacity which may explain the reduction in mtDNA copy number in patient fibroblasts”. It is confusing to compare two different observations in two different cell types. I suggest not to compare the observations directly. Besides, not R182W PolG2 showed significantly impaired respiratory capacity, but HEK293 cells showed significantly impaired respiratory capacity. Write, for instance: “HEK293 cells which expressed R182W PolG2 showed significantly impaired respiratory capacity, while patient fibroblasts showed a reduction in mtDNA copy number”.
Bottom Table 1: “*Genes in pink encode minimal mtDNA replication machinery”. No genes are coloured pink. Please explain.
Author Response
Reviewer 1:
- Reviewer comment: Page 8: “Symptoms from deletions usually don’t occur until >50% of the mtDNA is depleted while adult-onset symptoms from multiple DNA deletions are associated with deletions in >60% of the mtDNA genomes (91). Should this be: “Symptoms from depletions usually don’t occur until >50% of the mtDNA is depleted, while…”? Please check.
Authors’ response: The authors agree with the reviewer and have corrected the sentence as suggested by the reviewer. The revised sentence can be found on page 8 of the revised manuscript.
- Reviewer comment: Please explain the abbreviation “SEC-MALS”.
Authors’ response: The authors agree with the reviewer and have included the explanation for the SEC-MALS acronym in the revised manuscript on page 11 as well as in the list of abbreviations on pages 21-22.
- Reviewer comment: Page 10: “R182W PolG2 expressed in HEK293 cells showed significantly impaired respiratory capacity which may explain the reduction in mtDNA copy number in patient fibroblasts”. It is confusing to compare the observations directly. Besides, not R182W PolG2 showed significantly impaired respiratory capacity, but HEK293 cells showed significantly impaired respiratory capacity. Write, for instance: “HEK293 cells which expressed R182W PolG2 showed significantly impaired respiratory capacity, while patient fibroblasts showed a reduction in mtDNA copy number”.
Authors’ response: The authors agree with the reviewer and have corrected the sentence as suggested by the reviewer. The revised sentence can be found on page 10.
- Reviewer comment: Bottom Table 1: “Genes in pink encode minimal mtDNA replication machinery”. No genes are colored pink. Please explain.
Authors’ response: Given that the original formatting of the table was lost, the authors have omitted this statement from the final draft.
Reviewer 2 Report
Comments and Suggestions for Authors
The paper titled "Mitochondrial DNA Replication and Disease: A Historical Perspective on Molecular Insights and Therapeutic Advances" focuses on mitochondrial disorders over the past decade. The review is well conceived and well written. It is recommended that Figure 1 be revised to include all genes on both DNA strands. In Section 2, "DNA Polymerase," the current text should be reconsidered and replaced with content incorporating examples derived from human studies. Relevant references should be added to support the discussion.
Overall, the manuscript is well constructed and well written. It is recommended for acceptance after minor revisions.
Additional comments have been indicated in the PDF file. Please refer to the attached annotated PDF for further details.

Overall, the manuscript is well constructed and well written.
Author Response
Point by Point Response:
Reviewer 2:
- Reviewer comment: It is recommended that Figure 1 be revised to include all genes on both DNA strands.
Authors’ response: The authors agree with the reviewer and have updated Figure 1A to include the single letter abbreviations of all the tRNAs and outlined which genes are encoded on each strand.
- Reviewer comment: In Section 2, “DNA Polymerase g”, the current text should be reconsidered and replaced with content incorporating examples derived from human studies. Relevant references should be added to support the discussion. Authors’ response: The authors thank the reviewer for their careful reading of this manuscript. The authors specified examples from human as well as mammalian studies and have included the necessary references pertaining this section in the revised manuscript.
- Reviewer comment: Reviewer 2 also suggested several grammatical edits. Authors’ response: The authors thank the reviewer for the suggested grammatical edits. The authors have incorporated most of the grammatical edits suggested by the reviewer and have made some additional All changes made are highlighted in yellow in the revised manuscript. Some additional changes that we made are described below:
- In front of the author correspondence, the authors changed the (*) symbol to (#) in order to be consistent with the sign after Dr. William Copeland’s name in the authors list.
- In the last sentence of the Abstract on page 1, the reviewer suggested replacing “Here” with “In this review”. The authors have included the suggested edit and added “explore” instead of “review” in this sentence to avoid repetition of the word “review” in the same sentence.
- The reviewer added the abbreviation “MIRAS” on page 8. Therefore, the authors included this abbreviation in the list of abbreviations on pages 21-22.
- On page 12, the reviewer suggested replacing “control” with “as”. The authors did not include this suggested edit, given that this suggested edit did not flow well with our line of thought in this sentence. The authors have highlighted “control” in green (in the revised manuscript), since we thought that this would better explain our line of thought.
- On page 16, the reviewer suggested multiple edits in the sentence: “Conversely, in a previous study led by our group, biochemical analyses of 20 Twinkle disease variants revealed that these variants had similar helicase activity as the wild type enzyme under optimized in vitro conditions (198)”. The authors included all the suggested edits from the reviewer and had to remove “revealed that” from this sentence for a correct grammatical flow. The new sentence reads as follows: “In contrast, a previous study from our group demonstrated that 20 Twinkle disease variants have similar helicase activity as the wild type enzyme under optimized in vitro conditions (198)”.
- On page 16, in the second sentence of the first paragraph under “Other proteins involved with mtDNA maintenance” the reviewer suggested replacing “paper” with “review”. The authors have included the suggested edit and added “discussed” instead of “review” in this sentence to avoid repetition of the word “review” in the same sentence.
- On page 18, in the first sentence of the paragraph describing TFAM, the reviewer suggested changing “is responsible for facilitating the compaction of mtDNA” to “facilitates mtDNA compaction”. The authors have included this change and had to change the entire sentence to the present tense to properly reflect the incorporated suggestion from the reviewer.